# Evaluating the effect of a community score card among pregnant and breastfeeding women living with HIV in two districts in Malawi

Megan Kays[1]*, Godfrey Woelk[1], Tegan Callahan[2], Leila Katirayi[1], Michele Montandon[2], Felluna Chauwa[3], Anne Laterra[4], Veena Sampathkumar[3], Dumbani Kayira[5], Thokozani Kalua[6], Erin Kazemi[1], Heather Hoffman[7], Surbhi Modi[2]

1 Elizabeth Glaser Pediatric AIDS Foundation, Washington, DC, United States of America, 2 Centers for Disease Control and Prevention, Atlanta, Georgia, United States of America, 3 Elizabeth Glaser Pediatric AIDS Foundation, Lilongwe, Malawi, 4 Cooperative for Assistance and Relief Everywhere, Inc., Atlanta, Georgia, United States of America, 5 Centers for Disease Control and Prevention, Lilongwe, Malawi, 6 Malawi Ministry of Health, Lilongwe, Malawi, 7 Department of Biostatistics and Bioinformatics, Milken Institute of School Public Health, The George Washington University, Washington, DC, United States of America

* meg.kays@gmail.com

**Data Availability Statement:** All relevant data are within the paper and its Supporting information files.

## Abstract

Malawi faces challenges with retaining women in prevention of mother-to-child HIV transmission (PMTCT) services. We evaluated Cooperative for Assistance and Relief Everywhere, Inc. (CARE's) community score card (CSC) in 11 purposively selected health facilities, assessing the effect on: (1) retention in PMTCT services, (2) uptake of early infant diagnosis (EID), (3) collective efficacy among clients, and (4) self-efficacy among health care workers (HCWs) in delivering quality services. The CSC is a participatory community approach. In this study, HCWs and PMTCT clients identified issues impacting PMTCT service quality and uptake and implemented actions for improvement. A mixed-methods, pre- and post-intervention design was used to evaluate the intervention. We abstracted routine clinical data on retention in PMTCT services for HIV-positive clients attending their first antenatal care visit and EID uptake for their infants for 8-month periods before and after implementation. To assess collective efficacy and self-efficacy, we administered questionnaires and conducted focus group discussions (FGDs) pre- and post-intervention with PMTCT clients recruited from CSC participants, and HCWs providing HIV care from facilities. Retention of HIV-positive women in PMTCT services at three and six months and EID uptake was not significantly different pre- and post-implementation. For the clients, the collective efficacy scale average improved significantly post-intervention, (p = 0.003). HCW self-efficacy scale average did not improve. Results from the FGDs highlighted a strengthened relationship between HCWs and PMTCT clients, with clients reporting increased satisfaction with services. However, the data indicated continued challenges with stigma and fear of disclosure. While CSC may foster mutual trust and respect between HCWs and PMTCT clients, we did not find it improved PMTCT retention or EID uptake within the short duration of the study period. More research is needed on ways to improve service quality and decrease

**Funding:** This work has been supported by the US President's Emergency Plan for AIDS Relief (PEPFAR) through the United States Centers for Disease Control and Prevention (CDC) under the terms of grant number: GH000985.

**Competing interests:** The authors declare that they have no competing interests.

stigmatized behaviors, such as HIV testing and treatment services, as well as the longer-term impacts of interventions like the CSC on clinical outcomes.

## Introduction

There has been significant progress in prevention of mother-to-child HIV transmission (PMTCT) in Malawi; from 2010 to 2018, the number of new HIV infections in children 0–14 years declined from more than 15,000 to under 3,500 [1]. However, further progress must be made to reach elimination of mother-to-child HIV transmission (MTCT). The MTCT rate at the end of breastfeeding was estimated to be 7.8% in Malawi in 2018 [2].

Malawi faces ongoing challenges with retention of mothers and infants in PMTCT care. A study completed after the national roll-out of universal lifelong antiretroviral therapy (ART) for pregnant and breastfeeding women found that 17% of women were lost to follow-up within six months of initiating ART [3]. Further, women who initiated ART during pregnancy were five times more likely than women who were initiated on ART when not pregnant to never return to the health facility [3]. Additionally, only 75% of HIV-exposed infants (HEIs) received a DNA PCR test for early infant diagnosis (EID) by two months of age in 2018 [4], highlighting significant losses to follow-up of mothers and infants during the breastfeeding period.

Studies examining PMTCT retention among pregnant and breastfeeding mothers on life-long ART in Malawi demonstrate several common barriers including low male involvement in PMTCT, fear of disclosure to one's partner, stigma, and poor interactions with some health care workers (HCWs) [5–10]. Studies have found that improved quality of care, including *perceived* quality of care, improves health-seeking behaviors, retention in care, and ART adherence [11–13]. One means of addressing persistent and modifiable barriers to quality care is using social accountability approaches to empower clients and communities [14,15].

In the health sector, social accountability approaches are participatory practices that reinforce the complementary responsibilities and shared accountability of citizens, HCWs, and governments in the provision of quality health services. They empower service users to voice their needs and concerns, hold service providers and government to account, and create space for collective action to improve public sector performance. One social accountability approach is the community score card (CSC) © developed by Cooperative for Assistance and Relief Everywhere, Inc. (CARE). The CSC is a social accountability approach which uses a two-way, ongoing participatory approach for assessment, planning, monitoring, and evaluating public services. The CSC brings together the demand side (i.e., 'client') and the supply side (i.e., 'HCWs') of a service or program to jointly analyze issues underlying service delivery problems, find a common and shared way of addressing those issues, and continuously track commitments toward the implementation of those solutions in an ongoing process of quality improvement [15].

A review of eight CSC projects in five countries found evidence suggesting increased citizen empowerment, service provider and power-holder effectiveness, and improvements in services [16]. A cluster randomized controlled evaluation of the CSC's effectiveness in improving reproductive health-related outcomes in Malawi found that clinic catchment areas implementing the CSC had higher rates of home visits, increased postnatal care, and improved service satisfaction compared to control areas [17].

The CSC approach has shown promising results in maternal, neonatal, child, and other health services domains, but prior to this study it had not been used in an HIV service delivery

setting or to assess patient outcomes, such as retention in care. The US Centers for Disease Control and Prevention (CDC), CARE, the Malawi Ministry of Health (MOH), and the Elizabeth Glaser Pediatric AIDS Foundation (EGPAF) collaborated to adapt, implement, and evaluate the CSC approach for PMTCT services at 11 facilities in the Dedza and Ntcheu districts of Malawi. In 2015–2016, Malawi's HIV prevalence overall was 10.6%, but varied by region [18]. The Ntcheu and Dedza districts of Malawi's central region were priority districts within the national HIV response, and EGPAF supported the MOH facilities as an implementing partner. The average proportion of HIV-positive pregnant women retained at six months in these districts was 75%, with 64–71% of HEIs returning at six weeks for EID.

The adaptation and implementation of the CSC within PMTCT settings aimed to improve clinical outcomes for HIV-positive pregnant and breastfeeding women and their families by strengthening community engagement in HIV service delivery. In this paper, we assess the effect of the CSC on women's retention in PMTCT services, EID uptake, collective efficacy among the clients (the extent to which the women believed that the community could work together to improve care), and self-efficacy among HCWs in delivering quality services and community participation.

## Methods

### Intervention description

The CSC approach is a participatory forum that consists of five core phases (illustrated in Fig 1 and described in detail elsewhere [16,17]), four of which are repeated on a regular basis (called "rounds"), during which service users and service providers separately identify issues and then jointly propose solutions and monitor implementation of the solutions. For this project, CARE-Malawi staff implemented three rounds of issue identification and implementation periods of four months duration from September 2017-September 2018. This project adapted elements of the CSC approach originally aimed at broad-based community engagement to focus on engaging HCWs and clients of PMTCT clinical services to identify and solve PMTCT-related issues. The adaptation process and lessons learned are described elsewhere [19]. HCWs were

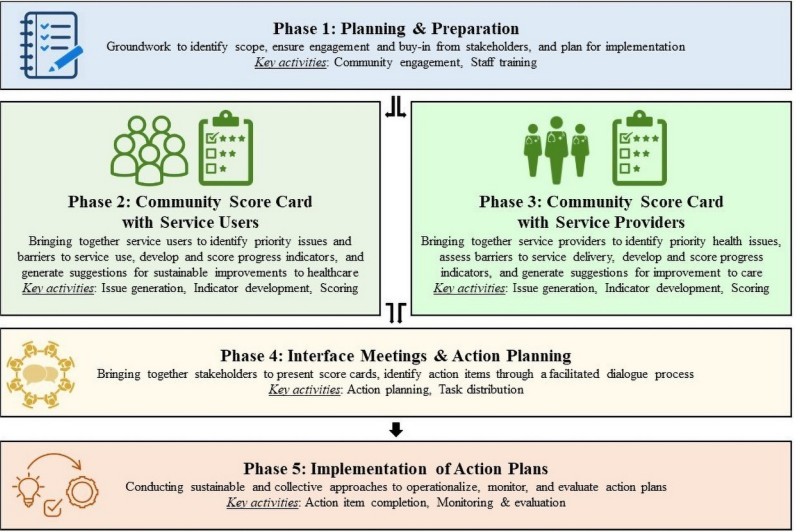

**Fig 1. Core phases of the community score card approach.**

recruited from the intervention clinics and PMTCT clients were identified through the intervention clinics and associations of people living with HIV within the catchment areas surrounding the intervention clinics and invited to participate in the CSC intervention. For the first round, HCWs and PMTCT clients separately identified the issues and barriers most impacting their ability to successfully deliver and access quality PMTCT services, then grouped these issues into indicators. These indicators were then scored by both HCWs and PMTCT clients and used to develop the individual score cards. HCWs and PMTCT clients then came together with other community and government leaders to present their scores and develop, implement, and monitor specific actions for improving PMTCT service delivery and access. Examples of specific actions identified were increasing the availability of trained HCWs providing PMTCT services and adequate infrastructure, supplies, and equipment. In subsequent rounds, previous issues were revisited in order to assess progress made as a result of the action period and re-scored, and action plans were updated. In the CSC approach, participants' *perception* of the quality of services meaningfully affects health-seeking behavior including decisions to remain in care [20,21]. A total of 822 PMTCT clients and 64 HCWs participated in the intervention, with participation varying by round.

## Study setting

The 11 health facilities in Ntcheu and Dedza districts (9 health centers and 2 district hospitals), were selected based on several criteria, including: 1) at least 25 HIV-positive women receiving HIV services annually; 2) less than 85% retention of HIV-positive pregnant and breastfeeding women in PMTCT services at six months after ART initiation; and 3) lower than national average uptake of EID. Ntcheu and Dedza district hospitals were included to test the CSC intervention in a setting with high patient volume.

## Study design

We used a mixed methods, pre- and post-intervention cross-sectional design to assess four key outcomes: (1) retention in PMTCT services, (2) uptake of EID, (3) collective efficacy among clients, and (4) self-efficacy among HCWs in delivering quality services.

To assess the effect of the CSC intervention on clinical outcomes (early retention in PMTCT for newly and previously diagnosed HIV-positive pregnant women and EID uptake), we abstracted data from routine health records maintained by the clinics in the study. The study used clinical records to determine retention in PMTCT services for HIV-positive pregnant and breastfeeding women and EID uptake for their HEIs in the periods before and after implementation of the intervention. Women did not have to participate in the CSC intervention to be included in the abstraction, as it was hypothesized that CSC-related clinical improvements may affect the quality of care, PMTCT retention, and EID uptake for all PMTCT clients.

To assess changes in collective efficacy and self-efficacy among the women and HCWs, we administered quantitative questionnaires and conducted pre- and post-intervention focus group discussions (FGDs) with PMTCT clients and HCWs engaged with the CSC intervention. We selected this mixed methods approach in recognition of the complex theories of change behind social accountability interventions [22]. Each component was conducted to provide unique information on the effectiveness of the intervention. By examining these findings together in one paper, we attempt a complete examination and discussion of the entirety of the findings to better understand the complex model of change.

## Study participants and sample size

**Clinical data abstraction.** In accordance with the MOH guidelines at the time of the study, pregnant women underwent "opt-out" HIV testing (default testing unless the woman refused testing) at first antenatal care (ANC) visit or delivery, and were immediately offered ART if found to be HIV-positive [23]. HIV-positive pregnant or breastfeeding women obtained monthly ART refills from maternal and child health or HIV clinics. HEIs were offered DNA PCR testing at 6–8 weeks of age.

For the abstraction of data from existing clinical records, women were eligible for inclusion if they were HIV-positive (newly or previously diagnosed), aged 15 years or older, and attended their first ANC appointment and received ART between August 2016-March 2017 (pre-intervention) and between August 2017-March 2018 (post-intervention) in one of the 11 selected facilities. All eligible women were identified using the clinic's ANC register and assigned a unique study identifier. All documented live births of the identified women were eligible for inclusion and listed with a linking identifier. We targeted a sample size of women and infants to detect a 10% change in six-month retention in PMTCT services and uptake of EID at 6–8 weeks, from baseline rates of 74% and 56% respectively. Given the hypothesized effect size, our power analysis determined a sample size of 481 women and 481 infants at pre- and post-intervention (power = 0.80, 2-tailed, $\alpha$ = 0.05, potential attrition = 20%) across the 11 sites was needed.

**Surveys and FGDs.** Surveys and FGDs were administered to CSC participants. PMTCT clients were eligible for inclusion if they were HIV-positive (newly or previously diagnosed), aged 15 years or older, and were currently pregnant or breastfeeding. All eligible PMTCT clients were approached and briefed about the survey at the first and last CSC meetings. Those who expressed interest were administered a consent script in a private place. Women who consented were invited to participate in the survey on the same day as the CSC meeting.

Participants for the FGDs were selected from CSC meetings using a lottery method. Attendees of the CSC meetings were provided with a number, and those who received numbers 2–6 were invited to participate in an FGD. Three FGDs with HCWs and three FGDs with the women were conducted in each of the two districts for the pre and post period. Those interested were given further details of the FGD. On the day of the FGD, eligibility criteria were reviewed for the women who came. Eligible women provided written consent.

HCWs were eligible for inclusion in the surveys and FGDs if they were 18 years and older and were working in HIV care in one of the selected facilities for at least one month. Eligible HCWs were recruited through the facilities. All eligible HCWs were invited to participate in the collective action survey and the FGDs. Booking for interviews and FGDs was conducted simultaneously, and consent was administered prior to each activity. With expected attendance of 300 PMTCT clients at CSC sessions and 65 eligible HCWs, we determined we would have at least 90% power to detect a 5% change in empowerment indicators in the survey. The subsets of these groups who participated in the FGDs were deemed sufficient in size to reach thematic saturation.

## Data collection

**Clinical data abstraction.** Trained research assistants abstracted clinical data from the ANC, ART, and maternity registers, as well as from the facility-maintained patient files documenting ART treatment for all eligible PMTCT clients. Information abstracted included socio-demographic characteristics, pregnancy information, birth outcomes, and visit information from up to seven ART visits recorded in the ART register. All HEIs of eligible PMTCT clients had their clinical information abstracted from the EID DNA PCR logbook and infant patient

files. Information abstracted included gender, birth weight, up to six months of visit information, and HIV testing and results. Abstraction for both populations was conducted from November 2017-January 2018 (pre-intervention) and October 2018-December 2019 (post-intervention). Research assistants recorded abstracted data on a paper form, which was then entered into an Epi Info v7.2 database.

Research assistants orally administered the surveys in Chichewa or English. The PMTCT client survey included questions on sociodemographic variables, collective efficacy, and at post-intervention only, participation in the CSC intervention. The HCW survey included questions on sociodemographic variables, work responsibilities, self-efficacy for collective action, social cohesion (mutual aid, trust, connectedness, and social support), work attachment, social participation, collective efficacy, and mutual responsibility. Seven research assistants were trained and responsible for conducting the FGDs in the role of moderator or notetaker. Research assistants recorded the responses on paper forms, which were then entered into an Epi Info database.

**Surveys and FGDs.** Research assistants facilitated the FGDs in Chichewa for PMTCT clients and English for HCWs. PMTCT client FGDs explored satisfaction with and perceived quality of health services, and relationship and trust with HCWs. HCW FGDs explored service challenges and accountability, and relationship and trust with clients. To reduce social desirability bias, the FGDs used anecdotal scenarios called "vignettes" to talk about the experience of new PMTCT clients and new HCWs. The vignettes focus on a young woman pregnant for the first time, attending her first ANC visit, and her experiences at the health facility, from her reception to her HIV-positive test, counseling, and drug pick-up. The scenario includes her perception of the services, and treatment by HCWs. At post-intervention, both populations were asked about the effect of the CSC on services. Prior to each discussion, FGD facilitators presented discussion guidelines, according to the standard operating procedures. In addition to assurances of the confidentiality of the discussions, the guidelines included the importance of confidentiality and the discouragement of sharing of personal information, especially for PMTCT clients. FGDs were audio recorded, transcribed, and for PMTCT client FGDs, translated to English.

## Ethical approval

Ethical approval was received from the Malawi National Health Sciences Research Committee, Advarra Institutional Review Board (IRB), and the CDC IRB. We obtained a waiver of informed consent for clinical data abstraction, as data was collected as part of routine care and it was not feasible to seek client permission. Informed consent was obtained from the clients and HCWs who participated in the questionnaire surveys and FGDs. In Malawi, pregnant women aged 15–17 years are emancipated minors and thus can provide informed consent for themselves and their infants. Clients provided written consent or a thumbprint if they were unable to sign their names, while HCWs provided written consent.

## Measures

We adapted validated measures from the "Women's and Health Worker Voices in Open, Inclusive Communities and Effective Spaces (VOICES): Measuring Governance Outcomes in Reproductive and Maternal Health Programmes" [24] to make them more explicitly related to PMTCT services. We adapted measures of collective efficacy for PMTCT clients, and self-efficacy, social cohesion, work attachment, collective efficacy, and mutual responsibility for and support of services for HCWs [25,26]. The modified collective efficacy scale consisted of four items: How sure are you that the community can work together to (1) obtain government

services and entitlements; (2) improve PMTCT services in this community; (3) improve how HIV -positive pregnant and postpartum women are treated at the health facility; and (4) improve the health and well-being of HIV-positive pregnant and postpartum women and their children. Collective efficacy was measured on a 5-point unipolar Likert scale from not at all sure to completely sure. The Cronbach alpha for the modified measure was 0.76, demonstrating sufficient internal consistency and scale reliability [27].

Self-efficacy for delivering quality health/PMTCT services was measured with two (modified) items: (1) how sure are you that you personally can do things to improve your own performance at work; and (2) how sure are you that you personally can do things to improve the quality of PMTCT services in the health facility and the catchment area. Self-efficacy for participation was measured with three items: (1) how sure are you that you can speak up in the community or health facility meetings about things that need improvement; (2) how sure are you that you can ask people in the community what HIV/PMTCT services their community needs; and (3) how sure are you that you can answer questions and share information with the community about PMTCT services available. All were measured on a 5-point unipolar Likert scale from not at all sure to completely sure. The Cronbach alpha for self-efficacy for delivering quality PMTCT services was 0.78 and 0.81 for participation.

Social participation was essentially an indicator of participation in the CSC process with questions on whether, in the past six months, they had met with the Village Health Committee, Health Center Committee, or community members to discuss and work on PMTCT issues. Participants were also asked if, in the past six months, they had meetings between the community, HCWs, and district government authorities where problems or other issues with health services were discussed, and plans for improving health services had been made to assess whether the collective action process had been implemented in full.

## Quantitative analysis

Abstracted clinical data were analyzed by calculating the percent of women retained in PMTCT services by timing of diagnosis (newly or previously diagnosed) and duration of time on ART (three or six months). Retention for this study was defined as having attended the most recent scheduled ART visit before the specified time (three or six months from first ANC), with a seven-day grace period. Chi square tests and logistic regression modeling were performed to examine the relationship between timing of diagnosis at both pre- and post-intervention and to compare PMTCT retention before and after the CSC intervention. We adjusted for type of health facility (i.e., hospitals or health centers) in the analysis. Similarly, the percent of HEIs tested with DNA PCR by eight weeks of age was calculated for pre- and post-intervention and compared. EID uptake was defined as having a recorded DNA PCR test in the infant's patient file or the EID DNA PCR logbook. We used generalized estimating equation (GEE) modeling to examine potential clustering. To account for the clustering of individuals within sites, we considered a compound symmetry working correlation structure in the GEE model. The correlation coefficient was negligible (close to 0), so we decided to proceed with logistic regression modeling and chi square tests. All statistical tests were conducted on the combined data for all sites and used $p < .05$ as the significance level.

The collective and self-efficacy measures were composed of items presented as questions that employed Likert scales with positively and negatively worded statements. These data were analyzed by summing the values for each item (a measure was composed of one or more items) and dividing by the number of items. For each measure, at both pre- and post-intervention, a mean and standard deviation (SD), and median and interquartile range (IQR), were determined. We assessed pre-/post-intervention differences using the Wilcoxon-Rank Sum

test, as we found that the data were not normally distributed. Analyses were conducted using STATA v12 and SAS v9.4.

## Qualitative analysis

FGDs were transcribed and translated into Word documents and uploaded into the qualitative analysis software program MaxQDA v12. A detailed code list was developed by the study team and used in the analysis of the pre- and post-intervention data. The pre-intervention data was analyzed after the first round of data collection, the post-intervention data was analyzed after the post-intervention data were collected, and then a comparison analysis was done to compare the results of the pre-/post-intervention data. This was done by comparing the data reduction and summary tables (matrices). The matrices compared the perspectives of the two populations at the two different time points. Data were compared pre- and post-intervention and changes in service quality, client satisfaction, and HCW-PMTCT client relationships were summarized, including specific changes attributed to the CSC intervention and process.

Data were coded by two research assistants for the pre-post analysis. For the comparison analysis, the study team reviewed the matrices and identified the changes pre-/post-intervention together. Reviews of coded transcripts were conducted throughout the coding process to ensure inter-coder reliability. This was done by comparing the coding of the same transcripts by the two research assistants.

## Results

### Clinical outcomes: Early retention of HIV-positive women enrolled in PMTCT and EID uptake

A total of 627 women at pre-intervention and 606 women at post-intervention enrolled in ANC were identified as eligible for data abstraction and had their records abstracted. All women had ANC information available, but ART and maternity information was not available for all women due to clients seeking health services at more than one health facility, poor documentation, and/or transfers, defaults, and deaths. At pre-intervention, 370 HEIs (59.0%) were identified for the 627 women, and at post-intervention, 448 HEIs (73.9%) were identified for the 606 women; not all HEIs were identified due to gaps in record documentation and linking systems.

Of the 627 HIV-positive women whose records were abstracted at pre-intervention, 43.4% (272) were newly diagnosed with HIV at their first ANC visit. At post-intervention, of the 606 HIV-positive women 38.6% (234) were newly diagnosed with HIV (Table 1). The mean ages of the women were 29 and 28 years at pre- and post-intervention, respectively. Women newly diagnosed with HIV tended to be younger compared to previously diagnosed women. Mean gestational age at first ANC visit was similar between women previously and newly diagnosed with HIV and both pre- and post-intervention (Table 1). A greater proportion of birth outcome data were available post-intervention.

Early retention of HIV-positive women enrolled in PMTCT at three and six months was not significantly different at pre- and post-implementation of the CSC intervention (Table 2). Of the 627 eligible women in the pre-intervention cohort, 476 (75.9%) and 475 (75.8%) had data available for 3-month and 6-month retention, respectively; out of 606 eligible women in the post-intervention cohort, 574 (94.7%) and 569 (93.9%) had data available for 3-month and 6-month retention, respectively. Pre-intervention, 362/476 (76.1%) HIV-positive women enrolled in PMTCT were retained at three months, while post-intervention, 446/574 (77.7%) were retained at three months, p = 0.527. At six months, 352/475 (74.1%) were retained

**Table 1. Characteristics of the women attending PMTCT and birth outcomes pre- and post- intervention.**

| | Pre-intervention | | | Post-intervention | | |
|---|---|---|---|---|---|---|
| | New HIV+ *(n = 272)* | Previous HIV+ *(n = 355)* | Total *(n = 627)* | New HIV+ *(n = 234)* | Previous HIV+ *(n = 372)* | Total *(n = 606)* |
| | N (%) | N (%) | N (%) | N (%) | N (%) | N (%) |
| **Age at first ANC visit (years)** | | | | | | |
| 15–19 | 35 (12.9) | 10 (2.8) | 45 (7.2) | 29 (12.4) | 13 (3.5) | 42 (6.9) |
| 20–24 | 86 (31.6) | 67 (18.9) | 153 (24.4) | 79 (33.8) | 67 (18.0) | 146 (24.1) |
| 25–29 | 64 (23.5) | 79 (22.3) | 143 (22.8) | 63 (26.9) | 98 (26.3) | 161 (26.6) |
| 30–34 | 59 (21.7) | 102 (28.7) | 161 (25.7) | 38 (16.2) | 95 (25.5) | 133 (21.9) |
| 35+ | 28 (10.3) | 97 (27.3) | 125 (19.9) | 25 (10.7) | 99 (26.6) | 124 (20.5) |
| Total | 272 (100.0) | 355 (100.0) | 627 (100.0) | 234 (100.0) | 372 (100.0) | 606 (100.0) |
| **Gestational age at first ANC visit (weeks)** | | | | | | |
| Mean (SD) | 20.7 (6.6) | 20.6 (6.7) | 20.7 (6.7) | 21.9 (6.2) | 21.8 (6.5) | 21.8 (6.3) |
| **Birth outcomes** | | | | | | |
| Alive | 147 (54.0) | 223 (62.8) | 370 (59.0) | 146 (62.4) | 302 (81.2) | 448 (73.9) |
| Still birth/ perinatal death | 0 (0.0) | 0 (0.0) | 0 (0.0) | 1 (0.4) | 1 (0.3) | 2 (0.3) |
| Missing | 125 (46.0) | 132 (37.2) | 257 (41.0) | 87 (37.2) | 69 (18.5) | 156 (25.7) |
| Total | 272 (100.0) | 355 (100.0) | 627 (100.0) | 234 (100.0) | 372 (100.0) | 606 (100.0) |

pre-intervention, and 431/569 (75.8%), post-intervention, p = 0.542. At both time periods, previously diagnosed HIV-positive women were more likely to be retained compared to newly diagnosed women (at 3 months, relative risk [RR] 3.73, 95% confidence interval [CI] 2.75–5.05; at 6 months RR 3.50, 95% CI 2.83–5.12).

Complete HIV testing was available for 326/370 (88.1%) of HEIs in the pre-intervention cohort and out of 419/480 (87.3%) HEIs in the post-intervention cohort. No significant differences in EID uptake were observed pre- and post-intervention, with EID uptake by eight weeks of age at 78.8% (257/326) pre-intervention and 77.6% (325/419) post-intervention, p = 0.678 (Table 2).

### Collective action surveys

A total of 596 PMTCT clients (295 pre-intervention and 300 post-intervention) and 128 HCWs (64 pre- and 64 post-intervention) participated in the collective action surveys. The

**Table 2. Number and percentage of HIV-positive pregnant women (newly and previously diagnosed) retained in PMTCT services at 3 and 6 months after their first ANC visit and infant DNA/PCR test at 6–8 weeks or >8 weeks of age, pre- and post-intervention.**

| | Pre-intervention | | | Post-intervention | | | P-value pre/post intervention |
|---|---|---|---|---|---|---|---|
| PMTCT retention[1] period | New HIV+ | Prev. HIV+ | Total | New HIV+ | Prev. HIV+ | Total | |
| | N (%) | N (%) | N (%) | N (%) | N (%) | N (%) | |
| 3 mo. | 126 (61.5) | 236 (87.1) | 362 (76.1) | 146 (65.0) | 300 (86.0) | 446 (77.7) | 0.527 |
| 6 mo. | 118 (57.8) | 234 (86.4) | 352 (74.1) | 140 (62.8) | 291 (84.1) | 431 (75.8) | 0.542 |
| DNA/PCR test[2] | N (%) | | | N (%) | | | 0.678 |
| 6–8 weeks | 257 (78.8) | | | 325 (77.6) | | | |
| >8 weeks | 69 (21.2) | | | 94 (22.4) | | | |

[1]Retention in PMTCT services is defined as having attended a clinic visit before the specified time (3- or 6-months after first ANC visit) *and* having a scheduled clinic visit after the specified time. A seven-day window was given to account for individuals with a clinic visit within one week before or after. Adjustments and exclusions were made to account for transfers and deaths. We excluded those for whom ART visit information was missing in pre- (n = 145) and post-intervention (n = 21) data sets.

[2]Excluding missing data; pre-intervention (n = 40) and post-intervention (n = 27).

**Table 3. Characteristics of PMTCT clients completing collective action surveys, pre- and post- intervention.**

| | Pre-intervention (n = 295) | Post-intervention (n = 300) |
|---|---|---|
| | N (%) | N (%) |
| **Age (years)** | | |
| 15–19 | 8 (2.7) | 4 (1.3) |
| 20–24 | 46 (15.6) | 47 (15.7) |
| 25–29 | 82 (27.8) | 71 (23.7) |
| 30–34 | 73 (24.8) | 76 (25.3) |
| 35+ | 86 (29.2) | 102 (34.0) |
| **Education level** | | |
| None | 41 (13.9) | 48 (16.0) |
| Primary | 194 (65.7) | 194 (64.7) |
| Secondary | 60 (20.3) | 58 (19.3) |
| **Time since HIV diagnosis** | | |
| Mean, SD | 4.3, 3.9 | 5.1, 4.0 |

characteristics of the PMTCT clients at pre- and post-intervention were similar, as shown in Table 3. The mean age and SD of the PMTCT clients were similar pre- and post-intervention at ages 30 (6) and 30 (7) years respectively. Similar proportions of clients attended primary school or less; 79.7% (235/295), and 80.7% (242/300) respectively. The mean time (SD) since HIV diagnosis was 4.3 (3.9) months for the pre-intervention clients and 5.1 (4.0) months for the post-intervention clients. Newly and previously diagnosed PMTCT clients' demographics were also similar.

Most of the HCWs interviewed were nurses or midwives (39.1% and 37.5% pre- and post-intervention, respectively). Most of the providers were from health centers (78.1% at pre- and post-intervention), as shown in Table 4. The majority had graduated more than four years prior and had at least 2 years of experience in their current role (data not shown).

Table 5 notes measures for collective and self-efficacy. The collective efficacy scale average demonstrated significant improvements for PMTCT clients at pre-/post-intervention, (p = 0.003). HCW self-efficacy scale average for delivering quality services and participation were high at baseline and did not improve post-intervention, p = 0.91 vs. p = 0.87.

**Table 4. Characteristics of HCWs completing collective action surveys, pre- and post- intervention.**

| | Pre-intervention (n = 64) | Post-intervention (n = 64) |
|---|---|---|
| | N (%) | N (%) |
| **Type of health facility** | | |
| Health center | 50 (78.1) | 50 (78.1) |
| Hospital | 14 (21.9) | 14 (21.9) |
| **Type of health worker*** | | |
| Doctor/clinical officer | 3 (4.7) | 3 (4.7) |
| Nurse/midwife | 25 (39.1) | 24 (37.5) |
| Client attendant | 1 (1.6) | 1 (1.6) |
| Health Surveillance Assistant | 13 (20.3) | 9 (14.1) |
| Other[#] | 21 (32.8) | 27 (42.2) |

*At pre-intervention, data on HCW type was missing for 1 participant.

[#] Medical assistant, nurse midwife technician, ART clerk.

**Table 5. Medians, interquartile ranges, and p-values for the measures of collective efficacy for the PMTCT clients, and self-efficacy delivering quality health service for participation for HCWs, pre and post intervention.**

|  | PMTCT clients | | | HCWs | | |
|---|---|---|---|---|---|---|
|  | Pre-intervention (n = 295) | Post-intervention (n = 300) | p-value | Pre-intervention (n = 64) | Post-intervention (n = 64) | p-value |
| Collective efficacy | 4.50 (4.0, 5.0) | 4.75 (4.25, 5.0) | 0.003 |  |  |  |
| Self-efficacy delivering quality health services | N/A | N/A |  | 4.00 (4.00, 4.00) | 4.00 (4.00, 4.00) | 0.91 |
| Self-efficacy for participation | N/A | N/A |  | 4.00 (3.67, 4.00) | 4.00 (3.67, 4.00) | 0.87 |

No significant changes were observed for HCWs at pre-/post-intervention for work responsibilities, social cohesion, work attachment, collective efficacy, and mutual responsibility (not shown; p>0.05). Social participation, however, did increase significantly at post-intervention (not shown; p = 0.021), as expected given the participation in the CSC intervention.

### FGDs

A total of 126 PMTCT clients participated in 12 FGDs (six pre- and six post-intervention) and 135 HCWs participated in 12 FGDs (six pre- and six post-intervention). Tables 6 and 7 present the demographic characteristic of the PMTCT and HCW participants in the FGDs, respectively. Women in the post-intervention FGDs were older, reported more primary education, and were more likely to be divorced, widowed, or separated compared to the pre-intervention group.

Compared to the pre-intervention FGDs, there was a higher proportion of female and hospital based HCWs in the post-intervention FGDs.

Results from the FGDs highlighted a strengthened relationship among HCWs and PMTCT clients, with many clients reporting an increased satisfaction in services after implementation of the CSC intervention.

**Table 6. Characteristics of participants in PMTCT FGDs, pre- and post- intervention.**

|  | Pre-intervention (n = 57) | Post-intervention (n = 69) |
|---|---|---|
|  | N (%) | N (%) |
| **Age (years)** |  |  |
| 15–19 | 7 (12.3) | 3 (4.3) |
| 20–24 | 11 (19.3) | 12 (17.4) |
| 25–29 | 16 (28.1) | 14 (20.3) |
| 30–34 | 7 (12.3) | 18 (26.1) |
| 35+ | 16 (28.1) | 22 (31.9) |
| **Education** |  |  |
| None | 12 (21.1) | 6 (8.7) |
| Primary | 32 (56.1) | 50 (72.5) |
| Secondary | 13 (22.8) | 13 (18.8) |
| **Relationship status** |  |  |
| Married/live as married | 49 (86.0) | 50 (72.5) |
| Divorce/widowed/separated | 4 (7.0) | 17 (24.6) |
| Never married | 4 (7.0) | 2 (2.9) |

**Table 7. Characteristics of participants in HCW FGDs, pre- and post- intervention.**

|  | Pre-intervention (n = 70)* | Post-intervention (n = 65) |
|---|---|---|
|  | N (%) | N (%) |
| **Gender** |  |  |
| Male | 38 (59.4) | 32 (49.2) |
| Female | 26 (40.6) | 33 (50.8) |
| **Type of facility** |  |  |
| Hospital | 12 (18.8) | 16 (24.6) |
| Health center | 52 (80.2) | 49 (75.4) |

*Six participants in the pre-intervention FGD elected not to share their demographic information.

> *"It is true that there is now a great improvement on our relationship, because the meetings we had with CARE tried to establish causes of all misunderstandings we had previously, and how to address those misunderstandings."*

(PMTCT client, post-intervention).

A major finding from the FGDs was the reported improvement in clinic service provision and the HCW-PMTCT client relationship. During the pre-intervention FGDs, PMTCT clients reported that the quality of services and consultations were good, but also noted long wait times, insufficient staffing and clinic hours, lack of privacy for consultation, and stock outs of supplies. HCWs also raised issues with space and supplies, as well as long hours and challenging working conditions. One HCW stated,

> *"[There is a] shortage of equipment. . .there are few rooms to work in. . .and there is too much work every day. . .every day, it's busy, working up at night and day. . .and it's hard work."*

Further, in the pre-intervention data, both HCWs and PMTCT clients reported interactions that were sometimes unpleasant or uncomfortable. PMTCT clients described HCWs as "rude" and reported that they would yell at clients, especially if they attended ANC without their male partners. One PMTCT client explained prior to the intervention,

> *"I was yelled at uncontrollably for forgetting to go for a scale weigh before meeting the nurse during my third ANC visit. Uh the nurse got angry; I feel like she overreacted."*

HCWs felt that PMTCT clients were often dissatisfied and unappreciative of their work. One HCW stated, '*It is true that that community members don't appreciate what HCW do.*'
Following the CSC, both PMTCT clients and HCWs reported substantial improvements in their communication and relationship with each other. A PMTCT client noted,

> *"Previously, it was very common for the health care workers to yell at clients or patients but now we have seen a great improvement. Nowadays it is very rare to experience that."*

At post-intervention, HCWs reported a greater understanding of clients' feelings and their needs, and PMTCT clients became more aware of what services they could expect from the health facility and the challenges that the HCWs face in providing services. One HCW reported,

*". . .[CSC] has helped, because the HCWs they were thinking they were doing right, but after attending the CSC meetings, they saw that people were complaining, and they have seen their areas to improve, and we have seen the change."*

A PMTCT client explained,

*"Previously, we had a lot of concerns which lacked a platform where they would be addressed the same with health care workers. However, with the introduction of CSC we had that platform to express ourselves. During these meetings we were able to understand other challenges that exist in the system."*

A similar positive impression of the CSC process was reported by an HCW,

*"The process is good because we were able to see the gaps we have in offering the services based on the discussions we had. We were able to understand what community members expect from us, and also the community had a chance to know what we can offer based on what we have on the ground."*

This mutual understanding fostered respect and improved relationships between HCWs and clients. One PMTCT client expressed that following the start of the CSC intervention, HCWs have "*started respecting us just as we respect them on their duty.*" Many women also expressed improved respect and treatment of clients by HCWs. One PMTCT client described HCWs as "*no longer harsh*" and others reported feeling more welcome at the health facility.

Both HCWs and PMTCT clients also reported concrete actions that had been implemented as a result of the CSC to improve PMTCT services, such as changing ART pickup times to less busy clinic times to provide more privacy. HCWs and PMTCT clients also reported increased timeliness of HCWs and more flexibility in working hours.

PMTCT clients also noted changes resulting from the CSC intervention not only in the facilities but also in the communities, including pastors sharing correct information about HIV and ART with others. An unexpected result of the CSC expressed by multiple PMTCT clients was a felt sense of reduced discrimination towards people living with HIV within the community, which was attributed to others being more aware of the dangers of discrimination. A PMTCT client shared her experience,

*"This program should continue because those who were discriminating others in the homes, relatives and communities have. . . stopped, because from the meetings, as for me. . . I was discriminated and they [parents] would not carry my baby. But now discrimination has ended."*

However, reduction in stigma was not universally expressed. Fear of disclosure of one's HIV status to a partner or family member was mentioned in the pre- and post-intervention FGDs, although the fear was not mentioned as frequently in the post-intervention data.

## Discussion

This study used multiple methods to assess the effect of the CSC on collective efficacy among PMTCT clients, self-efficacy among HCWs for delivering quality services, and perceived quality of care as well as key clinical PMTCT indicators including retention in PMTCT services and EID uptake. Improvements were seen in collective efficacy among PMTCT clients and perceived quality of care among both clients and HCWs. However, no significant changes were observed in HCW self-efficacy or in clinical PMTCT indicators.

The quantitative results from the collective action surveys showed a significant increase in women's certainty that the community could work together to improve PMTCT services. The qualitative results corroborated these findings, indicating improvements in client satisfaction with services and HCW-client relationships, suggesting perceptions of improved quality of care. Overall, the CSC intervention was highly valued by HCWs and PMTCT clients as a mechanism for improving communications, identifying approaches to improve health services, and increasing social participation. Despite this, there appeared to be no change in the HCWs self-efficacy in delivering quality health services. This may be due to HCWs believing that they were already providing acceptable services.

Additionally, no change was observed in the key patient-level clinical outcome indicators of early maternal PMTCT retention and EID uptake that we hypothesized would improve based on improved perceived and actual PMTCT service quality. Retention was significantly higher among previously diagnosed clients compared to those who were newly diagnosed, consistent with other studies; this may be related to age as the previously diagnosed women were older, and comparatively younger women are less likely to be retained [28,29]. However, there was no significant effect of the intervention on either group.

One potential explanation for the lack of statistically significant improvement in these indicators is the shortened intervention period and window for follow-up. In actual practice, the CSC process typically includes multiple rounds with longer action periods between rounds over the course of multiple years. These longer implementation and action periods and additional rounds may be needed for clients, HCWs and governments to effectively mobilize and implement transformative solutions, particularly operational mechanisms and procedures at the facility-level, and for these to translate into changes in behaviors and practices. In addition, some clients in the abstraction sample reached their three- and six-month retention windows and recommended EID uptake prior to the completion of all action periods for the CSC. With a longer period to make changes and to measure outcomes, significant improvements may have been observed.

A second explanation as to why statistically significant improvements in clinical outcomes were not observed could be that some barriers to PMTCT retention and EID are not particularly well suited to social accountability approaches. While the CSC intervention addressed some of the known barriers to PMTCT retention, particularly those related to patient-centered elements of quality [7] (e.g., the poor relationship between HCWs and clients), it was not as successful or well-suited at addressing health system and individual-level barriers such as a lack of integrated care, self-stigma, and partner disclosure [30]. A systematic review and series of meta-analyses on the relationship between HIV-related stigma and health identified a link between self-stigma and lower ART adherence and social support [30]. Self-stigma and fear of stigmatization as expressed in the FGDs may have inhibited women from disclosing their status and accessing services. Few documented social accountability projects have specifically engaged stigmatized populations, so it is unclear how community-based approaches function for these populations or how well approaches based on collective action and voice, like the CSC approach, work to improve private health behaviors for stigmatized health conditions like HIV/AIDS [16].

Other community-level interventions aimed at improving service uptake have likewise shown limited impact on clinical indicators. The CSC intervention conducted on maternal and child health in Malawi [17] found that while the CSC improved HCWs' outreach and increased service satisfaction ratings among women, there was no improvement in women's self-report of early ANC uptake or sufficient ANC attendance. Similarly, a community health worker intervention for people living with HIV in Tanzania showed improvements in adherence but not in retention in an analysis of clinical data before and after the intervention; they

hypothesized that community outreach improved adherence for women who had already overcome economic and social barriers to HIV care but did not ably address retention barriers [31]. Social accountability approaches like the CSC have demonstrated that they are well-positioned to impact important patient-centered elements of service quality such as patient perception, satisfaction, and trust and communication between client and provider, that are often under-valued and neglected in quality-of-care frameworks. These elements are often within the direct control of individual HCWs, facilities, and, to some extent, community members, and so may be more amenable to change through a process like the CSC, particularly when implemented over just a short period of time. The valuable contributions of these approaches should not be dismissed, especially where efforts to improve patient-centered dynamics of care have fallen short. However, they may not be the ideal mechanisms to rapidly address systemic and interpersonal issues that are often critical to PMTCT retention and EID uptake, particularly in a context of stigma or other individual-level barriers.

## Limitations

We collected data in only 11 selected facilities in two districts of Malawi, and thus may not be generalizable to other facilities or communities. During the intervention period, government, nonprofit, and community efforts were already being undertaken to address issues in HIV care and treatment, including PMTCT retention and EID uptake. As such, by the time this project was launched, these indicators were already steadily improving in the selected facilities, which may have diluted the effect of the CSC intervention.

Record availability and completeness was another limitation, particularly for the pre-intervention cohort; medical record completeness is a focus of ongoing clinical quality improvement interventions, which may have resulted in improved data availability at post-intervention, though this was not a focus of the CSC. Retention data were not available for eligible women either due to patient transfer to another facility or if the clinical card was not complete. There were challenges in finding records for eligible HEIs, and of those with records, some did not have an HIV testing date and were thus excluded from the HIV testing analysis.

Further, as noted in the Discussion, the modified CSC intervention included fewer rounds over a shorter period. This decision was made to accommodate the PMTCT timeframe; however, this timeframe may be too short to adequately affect and observe changes in clinical indicators, particularly as much of the clinical data reflected a period concurrent with the CSC intervention.

There was also variability in CSC implementation by site, which may have affected the outcomes. Variations in implementation and CSC actions by site were captured but were not included as a factor in the analysis as the study was not powered to analyze results by site.

A potential limitation is that the views from the respondents in the post-intervention FGDs may be more complementary of the CSC than those who did not participate. However, respondents who did not return may not have participated fully in the CSC intervention, and thus may not have been able to share more informed insights.

## Conclusions

Overall, this study found that the CSC approach could be a productive way to foster mutual trust and respect between providers and beneficiaries, to strengthen HCW-client relationships, and to make meaningful improvements in service quality. More research is needed on how to best capture the results of these community-based interventions and the extent to which they are suited to address individual behaviors in the short-term, especially for often stigmatized behaviors such as HIV treatment and testing. Research is also needed on whether and how

quickly changes in the relationships between service providers and users and in the quality of services can be expected to lead to improved clinical outcomes.

## Supporting information

**S1 File. EID abstraction baseline.**
(CSV)

**S2 File. EID abstraction endline.**
(CSV)

**S3 File. HCW survey baseline.**
(CSV)

**S4 File. HCW survey endline.**
(CSV)

**S5 File. PMTCT and ART abstraction data endline.**
(CSV)

**S6 File. PMTCT and ART abstraction baseline.**
(CSV)

**S7 File. PMTCT client survey baseline.**
(CSV)

**S8 File. PMTCT client survey endline.**
(CSV)

## Acknowledgments

We thank the CARE Malawi team that made this work possible, and the EGPAF Malawi team for supporting the activities. We also wish to thank the Malawi Ministry of Health and the District Health Management Teams in Ntcheu and Dedza for their support. We also thank Laura Reynolds, EGPAF Washington, for her support of the analysis and manuscript process. Finally, we offer our sincere appreciation for the participants in the PMTCT CSC intervention, and the clients and health care workers that gave their time for the evaluation.

## Author Contributions

**Conceptualization:** Megan Kays, Godfrey Woelk, Tegan Callahan, Michele Montandon, Anne Laterra, Veena Sampathkumar, Dumbani Kayira, Thokozani Kalua, Surbhi Modi.

**Data curation:** Felluna Chauwa.

**Formal analysis:** Leila Katirayi, Erin Kazemi, Heather Hoffman.

**Methodology:** Megan Kays, Godfrey Woelk, Tegan Callahan, Leila Katirayi, Michele Montandon.

**Supervision:** Megan Kays, Godfrey Woelk, Felluna Chauwa, Veena Sampathkumar, Dumbani Kayira, Thokozani Kalua.

**Writing – original draft:** Megan Kays.

**Writing – review & editing:** Godfrey Woelk, Tegan Callahan, Michele Montandon, Anne Laterra, Surbhi Modi.

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
