## [Decision Letter · Decision Letter 0]

17 Feb 2021

PONE-D-20-35790

Evaluating a Community Score Card for Pregnant and Breastfeeding Women Living with HIV in Malawi

PLOS ONE

Dear Dr. Kays,

Thank you for submitting your manuscript to PLOS ONE. After careful consideration, we feel that it has merit but does not fully meet PLOS ONE’s publication criteria as it currently stands. Therefore, we invite you to submit a revised version of the manuscript that addresses the points raised during the review process.

We look forward to receiving your revised manuscript.

Kind regards,

Hannah Tappis, DrPH, MPH

Academic Editor

PLOS ONE

Journal Requirements:

2. Please include a copy of the interview guide used in the study, in both the original language and English, as Supporting Information, or include a citation if it has been published previously.

Reviewers' comments:

Reviewer's Responses to Questions

**Comments to the Author**

1. Is the manuscript technically sound, and do the data support the conclusions?

Reviewer #1: Yes

Reviewer #2: Yes

Reviewer #3: No

2. Has the statistical analysis been performed appropriately and rigorously? 

Reviewer #1: Yes

Reviewer #2: Yes

Reviewer #3: No

3. Have the authors made all data underlying the findings in their manuscript fully available?

Reviewer #1: Yes

Reviewer #2: Yes

Reviewer #3: No

4. Is the manuscript presented in an intelligible fashion and written in standard English?

Reviewer #1: Yes

Reviewer #2: Yes

Reviewer #3: No

5. Review Comments to the Author

Reviewer #1: General summary comments from Reviewer.

The paper is generally well written and structured. This is an evaluation of Community Score Card in improving several PMTCT outcomes like retention and EID uptake. The study presents interesting results pertaining to patient and health worker collaboration to achieve a common goal and this makes the work novel.

However, I observe that the paper has some shortcomings and not clear why the study took place in only the specified targeted districts since the data used to express the gap had been national level data e.g national level prevalence, retention rate, and EID uptake. It would have been great if the author could also demonstrate the gaps at district level. The Methods section did not clearly state the significance level of the study error margin (e.g. 0.05 as significance level). The statistical analysis in this paper was appropriate and for all the significant associations, the author clearly specified the correponding p-values. On the other hand, the author may also wish to be more systematic : define both the primary and secondary outcomes ; then univariate analysis ; bivariate analysis and then multivariate. In terms of the study broader objective, the author ought to be consistent throughout the document: to use assess rather that describe or evaluate.– the author has used the term ‘describe ’ and ‘Evaluate’, but it seems the author means to say ‘assess’ (on p. 6, line 74 and line 104).

sOne of the limitatios of the evaluation is the data gaps, in the absense of proper data management sysytem for an intervention, results from the evaluation may not give a true reflection of change attributed to a particular intervention.

Major Comments:

- The author clearly specifies the study design

- The actual implementation of the intervention was summarized in such a way that it is difficult for the reviewer to understand what actually happened on the ground. Additional information on actual implementation of CSC is required.

- The variables tracked in the score card are not specified, this includes the indicators of issues that were identified and also the actual scores during the baseline and thereafter. Inclusion of score card indicators could have set a good background of the evaluation.

- The author to be consistent in the overall aim of the study: either to use assess or describe or evaluate throughout the document. (on p.6; par 2; line 74 and on p.7; par 4; line104)

- The discussion section has been well articulated

- The paper should clearly indicate the data management system that was introduced during the actual implementation of CSC and the evaluation could have also identified the gaps and recommend proper data management system for CSC rather than included that as an area of research.

Minor Comments:

- The authors should also consider specifying the significance margin error in the methodology section (e.g at 0.05 as significance level)

- One sentence in the Data section must be revised to be grammatically correct(P.11; Line 184)

- Include the prevalence, retention rate, and EID uptake for the study districts in the introduction section. These will help justify why the two districts were included in the priority districts for HIV/AIDS interventions.

- It is not clear how informed consent for the minors (15-17 years) was done

Reviewer #2: The article provides interesting and useful results on the efficacy and methods of a potential intervention to improve prevention of HIV mother-to-child transmission services. Despite the lack of evidence for an impact on some of the outcomes the manuscript shares very useful information on some of the perceived challenges with the PMTCT services. The authors have written the manuscript clearly and provided the relevant data to understand the results.

I do not have any big picture concerns. The statistical methods are appropriate. The methods for handling the qualitative data are thoroughly described.

I have a number of editorial and clarifying comments.

1. The data visuals are not overly appealing for an audience. Consider revising Table 2 into a graphic to show the pre and post results. Even a non-significant finding will still capture the audience’s eye faster in a graphic than a table. (not required, just for consideration)

2. Line 144-145. This sentence was unclear to me. "A subset of women who participated in the survey were randomly approached using a number, n, 2-6 for a FGD that took place at a different date and venue." Please explain what is meant that the women were approached with a number. Perhaps they were randomly assigned a number?

3. Line 154. Should "consenting" be "consent"

4. Line 249: the authors suggest that the reviews of the coded transcripts were done throughout the process but then mention three distinct time points. It seems like the sentence should either say “throughout the process” or at the three time points, but not both. Also you could consider providing a sentence on what was done to ensure inter-rater reliability or at a minimum how many different team members were doing the coding.

5. Line 291: it was interesting how different retention results were for women already on ART vs women starting ART in this study vs the previous research noted in the start of the document. Please comment on that in the discussion.

6. Line 444: The reflection on the lack of engagement with stigmatized populations is an important comment in the discussion. No action is necessary, I am just commending the authors for including this very important point.

7. Lines 459-464. Should this concluding statement come after the limitations – so explaining what was found despite the limitations mentioned before?

8. To meet the data transparency and availability requirements, I believe the statement that data are available from EGPAF should be included within the text of the manuscript.

Reviewer #3: Comments

Congratulations to the authors on the very important paper. Eliminating MTCT remains a critical challenge and such studies are essential to identify and stimulate stakeholder participation, patient cantered care and hopefully the desirable health outcomes.

I have the following suggestions for the authors to consider in improving their work.

1. The title of the paper could benefit from including what the outcome is so readers know from the onset what is being measured and it also sets the stage for what is follow in the paper – what is the intention of the score card and what are you expecting to change among your target population?

2. Still on the title, the study was done in a couple of districts and not the entire Malawi, the authors can be more precise by naming the districts or simply saying “in two districts in Malawi”.

3. The abstract and body of the paper could benefit from inclusion of participant selection or sampling method and criteria

4. Were the facilities compared all at the same level and what is the implication of lumping together different level of care in one analysis – any risk for of bias?

5. Better description on the distribution of sample by level care (if any) may also help understand any nuances that may be confounders to adjust for.

6. Did you have a same or separate sample pre and post intervention?

7. Still on the intervention I wasn’t sure if you used the same cohort for the different rounds of the CSC and if that has implication for the outcome of the study?

8. I do not think that CSC is appropriate for some of the outcomes given what CSC is intended for and what it can not do. In this regard, the background could benefit from a richer description of what the impediments to PMTCT service outcome are and which of them are amenable to CSC and make the study about such outcome. It is unsurprising that it had no impact on retention or EID given that their known determinants are more than relationship and power interactions and also includes psychosocial and structural factors beyond the competence and scope of the health care workers. The lower rate of retention of newly diagnosed HIV clients is one example to show that what affects retention is beyond service delivery and access.

9. With the above I am uncomfortable with the mismatch of problem and CSC solution. I feel the paper can be better structured from start to finish focusing on what CSC can achieve given the preponderance of evidence of what drives various PMTCT outcomes.

10. The six aims of improvement suggested by the Institute of Medicine may help to position element of improvement CSC can address and should be focus of the paper, otherwise will be an unfair or an insufficiently considered assessment – see link for application of IOM’s six aims of improvement https://doi.org/10.1097/01.pcc.0000160592.87113.c6.

11. In terms of the FGDs, will post intervention FGD of retained client not be biased to the views of those who were retained and may be exaggerate the intervention’s benefits?

12. Who conducted the data abstraction and interviews?

13. How was confidentiality maintained in both instances particularly the FGD of people living with HIV?

14. Why did you need ethics wavier for data abstraction if you had access to participants for your intervention why did they not consent to your using their data?

15. Who conducted the intervention?

16. What kind of actions were implemented to address issues emerging from the CSC?

17. How were the actions tracked to see if they changed before assessing impact? If nothing changed, can it be attributed to the CSC or other elements?

18. It may help to have a table of outcomes, data collection tools and reliability of the different scales. It is hard to follow from the narrative.

19. The description of qualitative analysis method could be improved to demonstrate process and stages of analysis, by who, quality assurance and trustworthiness elements etc.

20. The mixed methods used approach should be specified given its implication for data analysis and presentation, I may be wrong but it appears to me as a concurrent triangulation and would have preferred to see a greater mixing or integration of both quantitative and qualitative data rather than the observed clear separation

21. Back to my point on fair use of CSC, the discussion acknowledges its inability to address psychosocial issues which is an essential and established requirement for retention and mostly why newly tested were less likely to be retained due to adjustment issues that are amenable to psychosocial support.

22. Given what know about PMTCT today this paper may be more novel if it focuses on its elements of collective efficacy and with better description of the intervention, study design and methodology.

6. PLOS authors have the option to publish the peer review history of their article (what does this mean?). If published, this will include your full peer review and any attached files.

Reviewer #1: **Yes: **Sam Phiri

Reviewer #2: No

Reviewer #3: No

---

## [Author Response · Author response to Decision Letter 0]

18 Apr 2021

Reviewer Comment Revision made

1 Not clear why the study took place in only the specified targeted districts since the data used to express the gap had been national level data e.g national level prevalence, retention rate, and EID uptake. It would have been great if the author could also demonstrate the gaps at district level. Include the prevalence, retention rate, and EID uptake for the study districts in the introduction section. These will help justify why the two districts were included in the priority districts for HIV/AIDS interventions. Thank you for this comment. We agree and have added some information in the Introduction on retention and EID uptake, as well as clarifying that these are intervention districts for EGPAF.

1 The Methods section did not clearly state the significance level of the study error margin (e.g. 0.05 as significance level). This is useful feedback. The p-value was (p<.05) was added to the Statistical Analysis section.

1 The author may also wish to be more systematic : define both the primary and secondary outcomes ; then univariate analysis ; bivariate analysis and then multivariate. Thank you for this suggestion. We feel the way the results are currently presented is the most straightforward, so we are maintaining this section as is.

1 In terms of the study broader objective, the author ought to be consistent throughout the document: to use assess rather that describe or evaluate.– the author has used the term ‘describe ’ and ‘Evaluate’, but it seems the author means to say ‘assess’ (on p. 6, line 74 and line 104). Thank you for this useful suggestion. We have changed the instances of “describe” and “evaluate” to “assess.”

1 The actual implementation of the intervention was summarized in such a way that it is difficult for the reviewer to understand what actually happened on the ground. Additional information on actual implementation of CSC is required. Thank you for this suggestion. We have added a few more clarifying details in the Intervention Description. We have also referenced the paper that was written on the adaptation that provides more details. 

1 The variables tracked in the score card are not specified, this includes the indicators of issues that were identified and also the actual scores during the baseline and thereafter. Inclusion of score card indicators could have set a good background of the evaluation. Thank you, we agreed with this suggestion and added a few examples of issues identified to the Intervention Description.

1 The paper should clearly indicate the data management system that was introduced during the actual implementation of CSC and the evaluation could have also identified the gaps and recommend proper data management system for CSC rather than included that as an area of research. Thank you for flagging that this was not clear. We have added some information at the start of the study design description to clarify that we used existing clinical records. 

1 One sentence in the Data section must be revised to be grammatically correct(P.11; Line 184) Thank you for the careful read. However, we did not find any grammatical inaccuracies in this section upon review.

1 It is not clear how informed consent for the minors (15-17 years) was done Thank you for noting this. We added this sentence in the Ethical Approval section: In Malawi, pregnant women aged 15-17 years are emancipated minors and thus can provide informed consent for themselves and their infants.

2 The data visuals are not overly appealing for an audience. Consider revising Table 2 into a graphic to show the pre and post results. Even a non-significant finding will still capture the audience’s eye faster in a graphic than a table. (not required, just for consideration) Thank you for your suggestion. However, we believe that the table efficiently sums up and presents the necessary information

2 Line 144-145. This sentence was unclear to me. "A subset of women who participated in the survey were randomly approached using a number, n, 2-6 for a FGD that took place at a different date and venue." Please explain what is meant that the women were approached with a number. Perhaps they were randomly assigned a number? Thank you for noting this. We have clarified this sentence to indicate that the women were randomly assigned a number.

2 Line 154. Should "consenting" be "consent" Thank you for this edit. We changed this to consent.

2 Line 249: the authors suggest that the reviews of the coded transcripts were done throughout the process but then mention three distinct time points. It seems like the sentence should either say “throughout the process” or at the three time points, but not both. Also you could consider providing a sentence on what was done to ensure inter-rater reliability or at a minimum how many different team members were doing the coding. Thank you for this suggestion. We have edited the Qualitative Analysis section to provide additional details on how the coding was conducted.

2 Line 291: it was interesting how different retention results were for women already on ART vs women starting ART in this study vs the previous research noted in the start of the document. Please comment on that in the discussion. Thank you for the interest. We have commented on this in the Discussion. The previous research compared retention of PMTCT women with non-pregnant women initiating ART. We did not make this comparison in our study. 

2 Lines 459-464. Should this concluding statement come after the limitations – so explaining what was found despite the limitations mentioned before? Thank you for this suggestion. We have added a Conclusions section following the limitations and have added this text to the Conclusions.

2 To meet the data transparency and availability requirements, I believe the statement that data are available from EGPAF should be included within the text of the manuscript. Thank you for noting our omission. We have added a statement on data transparency.

3 The title of the paper could benefit from including what the outcome is so readers know from the onset what is being measured and it also sets the stage for what is follow in the paper – what is the intention of the score card and what are you expecting to change among your target population? Thank you for this suggestion. We felt that simplicity in the title was preferred.

3 Still on the title, the study was done in a couple of districts and not the entire Malawi, the authors can be more precise by naming the districts or simply saying “in two districts in Malawi”. Thank you for this suggestion. We did add “in two districts in Malawi” to clarify the scope of the intervention.

3 The abstract and body of the paper could benefit from inclusion of participant selection or sampling method and criteria Thank you for this suggestion. We have added more sampling and inclusion information to the abstract. We have more clearly indicated the sampling method for women participating in the surveys and FGDs in the Methods section.

3 Were the facilities compared all at the same level and what is the implication of lumping together different level of care in one analysis – any risk for of bias? Better description on the distribution of sample by level care (if any) may also help understand any nuances that may be confounders to adjust for. Thank you for this question. We adjusted for the different types of facilities, (hospital/health centers) in the analysis, through the logistic regression, and have specified that in the Statistical Analysis section.

3 Did you have a same or separate sample pre and post intervention? Thank you for this question. This was not a cohort study, so have tried to clarify that the pre and post were based on time period. 

3 Still on the intervention I wasn’t sure if you used the same cohort for the different rounds of the CSC and if that has implication for the outcome of the study? Thank you for this question. We added language to Intervention Description to clarify that the participation in the CSC varied. We also added information to the Methods to further clarify that all women were eligible for abstraction, even if not participating in the CSC.

3 I do not think that CSC is appropriate for some of the outcomes given what CSC is intended for and what it can not do. In this regard, the background could benefit from a richer description of what the impediments to PMTCT service outcome are and which of them are amenable to CSC and make the study about such outcome. It is unsurprising that it had no impact on retention or EID given that their known determinants are more than relationship and power interactions and also includes psychosocial and structural factors beyond the competence and scope of the health care workers. The lower rate of retention of newly diagnosed HIV clients is one example to show that what affects retention is beyond service delivery and access. Thank you for this critique. We chose to test the CSC in a PMTCT setting as it has already been used for other health care issues and has been shown to be an effective approach for improving provider-client relationships, and thus perceived and actual quality of care. We do agree that this was not as clearly spelled out in this paper so we have added more information to the Introduction to clarify our use of the CSC, indicating that provider-client relationships can be a barrier to PMTCT and that CSC is a tool that can be used to improve these relationships, and thus clinical outcomes.

3 With the above I am uncomfortable with the mismatch of problem and CSC solution. I feel the paper can be better structured from start to finish focusing on what CSC can achieve given the preponderance of evidence of what drives various PMTCT outcomes. Again, thank you for flagging this. We have not changed the structure of the paper, as there is an accompanying paper that provides more information on the use of the CSC in this situation. However, we have added more information to the Introduction, as previously noted.

3 The six aims of improvement suggested by the Institute of Medicine may help to position element of improvement CSC can address and should be focus of the paper, otherwise will be an unfair or an insufficiently considered assessment – see link for application of IOM’s six aims of improvement https://doi.org/10.1097/01.pcc.0000160592.87113.c6.

Thank you for this resource. The accompanying CSC paper referenced uses the Kruk et al high quality health system framework, which is similar to IOM. We were testing whether this process could affect joint actions to address poor service quality. It is also well-suited to address patient-centeredness.

3 In terms of the FGDs, will post intervention FGD of retained client not be biased to the views of those who were retained and may be exaggerate the intervention’s benefits? Thank you for this question. We agree this is a potential limitation, and we have added the text to the Limitations section. 

3 Who conducted the data abstraction and interviews? Thank you for this request for clarification. We have added more text to make it clearer that the data collection activities were conducted by research assistants.

3 How was confidentiality maintained in both instances particularly the FGD of people living with HIV? Thank you for this suggestion. We agree and have added some information in Methods on how confidentiality was protected. 

3 Why did you need ethics wavier for data abstraction if you had access to participants for your intervention why did they not consent to your using their data? Thank you for this note. We added some language to clarify why consent was not sought in the Ethical Approval section.

3 Who conducted the intervention? Thank you for this question. We have clarified in the Intervention Description that the CSC was implemented by CARE-Malawi staff.

3 What kind of actions were implemented to address issues emerging from the CSC? Thank you for this suggestion. We agree and have added some examples to the Intervention Description. 

3 How were the actions tracked to see if they changed before assessing impact? If nothing changed, can it be attributed to the CSC or other elements? Thank you for this suggestion. We have included some information in the intervention description, and the existing qualitative results note perceived improvements.

3 It may help to have a table of outcomes, data collection tools and reliability of the different scales. It is hard to follow from the narrative. Thank you for this idea. We have included the scales used and their reliability in a table, replacing the text.

3 The description of qualitative analysis method could be improved to demonstrate process and stages of analysis, by who, quality assurance and trustworthiness elements etc. Thank you for this suggestion. We have added more details about how the analysis was conducted in the Qualitative Analysis section.

3 The mixed methods used approach should be specified given its implication for data analysis and presentation, I may be wrong but it appears to me as a concurrent triangulation and would have preferred to see a greater mixing or integration of both quantitative and qualitative data rather than the observed clear separation Thank you for this critique, and we appreciate this idea. In the design, these were unique components that were then brought together at the analysis phase to provide context and clarity. 

3 Given what know about PMTCT today this paper may be more novel if it focuses on its elements of collective efficacy and with better description of the intervention, study design and methodology. Thank you for this comment. Collective efficacy was one outcome of interest, but is not the only outcome associated with the CSC. We felt that the more novel aspect was a holistic evaluation of the intervention. Fewer studies in the literature have tried to link community action and clinical outcomes, which is one of the aspects we were assessing. As such, we have chosen to retain the focus of this paper as providing a broader view of the CSC’s performance.

---

## [Decision Letter · Decision Letter 1]

14 Jun 2021

PONE-D-20-35790R1

Evaluating the effect of a Community Score Card among pregnant and breastfeeding women living with HIV in two districts in Malawi

PLOS ONE

Dear Dr. Kays,

Thank you for submitting your manuscript to PLOS ONE. After careful consideration, we feel that it has merit but does not fully meet PLOS ONE’s publication criteria as it currently stands. Therefore, we invite you to submit a revised version of the manuscript that addresses the points raised during the review process.

We look forward to receiving your revised manuscript.

Kind regards,

Hannah Tappis

Academic Editor

PLOS ONE

Journal Requirements:

Reviewers' comments:

Reviewer's Responses to Questions

**Comments to the Author**

1. If the authors have adequately addressed your comments raised in a previous round of review and you feel that this manuscript is now acceptable for publication, you may indicate that here to bypass the “Comments to the Author” section, enter your conflict of interest statement in the “Confidential to Editor” section, and submit your "Accept" recommendation.

Reviewer #4: All comments have been addressed

Reviewer #5: (No Response)

2. Is the manuscript technically sound, and do the data support the conclusions?

Reviewer #4: Partly

Reviewer #5: Partly

3. Has the statistical analysis been performed appropriately and rigorously? 

Reviewer #4: I Don't Know

Reviewer #5: I Don't Know

4. Have the authors made all data underlying the findings in their manuscript fully available?

Reviewer #4: Yes

Reviewer #5: Yes

5. Is the manuscript presented in an intelligible fashion and written in standard English?

Reviewer #4: Yes

Reviewer #5: Yes

6. Review Comments to the Author

Reviewer #4: The original manuscript has undergone meticulous review by a previous reviewer on methodology, study design sampling etc and most of the issues and concerns raised have been addressed by the authors. Hence my observations are primarily on the CSC presentation, intervention, discussion, and interpretation of the results.

L2 The abstract could be improved by including the figure for low retention rate, and care continuum for PMTCT in Malawi. This may provide a stronger rationale for researching the complimentary CSC intervention.

Study design and selection of Health facilities and hospitals needs to be provided in the methods section of the abstract. How were the districts and facilities sampled? Stratified random sampling based on public/private facilities? Both district hospitals?

The key limitation of intervention duration may be mentioned in the conclusion.

L125 “Women did not have to participate in the CSC intervention to be included in the abstraction, as it was hypothesized that CSC-related clinical improvements may affect the quality of care, PMTCT retention, and EID uptake for all PMTCT clients” This sentence illustrating the selection of the sample, seems to be contrary to the principles of the CSC where engagement, ownership to the process leads to change, uptake of services or perceived quality due to visible improvements as illustrated by the scorecard

The authors may consider including a CSC model and describe the details of the process. References are provided but further explanation on the implementation of the generic model in the Malawian context is needed. How was the CSC introduced, were facilitated meetings scheduled for joint discussion and deliberation of results and the rationale for scoring poor, or high?

L169 Experience of HCW was as low as 1m? HCW has minimal time to engage with communities, due to the workload, shortages in staffing, and high priority services. It is unlikely that HCW who have 1m experience will be able to effectively address community needs, or challenges with the PMTCT demand side challenges without prior experience. This needs to be discussed as a limitation. Who were ‘other’ HCW cadre?

CARE has a well established CSC model that has been successfully adapted in other contexts. However, the period of intervention (1y?), maybe insufficient to establish the various community mechanisms and systems at the facility level to ensure optimal operations and internalization of the concept and procedures.

The CSC methodology needs more detailed description to improve the study rigor. The discussion or methodology does not include measures for implementation integrity and intensity of the CSC. Inclusive representation in the community CSC?

The facility aspects need to be optimized and integrated, as ART and maternity information was not available in one HF

L249. Level of engagement and contribution in meetings needs to be measured, rather than generic indicators for meeting participation.

Were the 606 women assessed in the post evaluation included in the 627 reported in the pre evaluation?

75% had data available in pre and 93-94% in post, was this improvement a result of the CSC?

How did the HCW participate in the CSC, what was their specific role and continuing responsibility, were HCW designated to facilitate follow up with the community? Were problem solving mechanisms or action plans instituted following the CSC mtgs? Was there an open community forum to discuss and deliberate results?

The authors have embarked on a critical people centered paradigm for enhancing health improvements using the CSC model. However, the mechanisms of implementation integrating an inclusive approach of patients and HCW need careful consideration and deliberation, for a meaningful interpretation of the effectiveness for health outcomes and health system improvements. Patient perception, satisfaction, perceived reduced discrimination must be weighted equally with the validated outputs/outcomes for the PMTCT utilization, continuum of care and quality, for a value based healthcare system. Harsh, disrespectful, unpleasant provider patient encounters need to be valued and integrated in quality of care framework.

Reviewer #5: Title: Evaluating the effect of a Community Score Card among pregnant and breastfeeding women living with HIV in two districts in Malawi.

Abstract: This is good

Introduction: This section is written well and summarized

Methods:

Intervention description

To be able to track improvement in service delivery

“Line 95-96- “Were the clients who were recruited utilizing services from the selected facilities? Where were these clinics and associations of people living with HIV selected from?

Line 97-99-Did the HCWs and PMTCT clients identify the issues and barriers most impacting their ability to successfully deliver and access quality of PMTCT services jointly or did HCWs do it alone and clients identified theirs alone? That part is not clear in the write-up

Line 104-105- What was the aim of revisiting previous issues?

Line 100-102-Was it only community leaders or community leaders and community members?

Line 165-167- I would think that for one facility, the quota should be 12 and not six as stated. Please clarify on this

Line 183 From what?

Line 184-6-Was ANC the point of identification of the clients who were included in the study?

Line 183-186- This sounds like a result which should be in the results section

Line 196-7-How do define social cohesion?

Line 201- How many research assistants facilitated the FGDs for PMTCT clients and how many research assistants facilitated FGDs for HCWs?

Line 275-What were the scores and How were the scores obtained. That is not explained

Results

Line 307- mean ages should be reported with corresponding standard deviations

Line 368-371- The table should include significant results

Line 381- was it a requirement for these participants not to share their socio demographic characteristics?

Line 382-3-It’s appropriate to report qualitative data in themes and sub themes and then you compare those themes pre- and post-intervention. This could be based on the different aspects that authors set out to evaluate e.g various aspects of PMTCT services

Line 384-386-It would be appropriate to first report the relationship that existed previously before the intervention was implemented and then the authors can report improved relationship after the project has been implemented

Line 393-4-Is it possible to have a quote for this?

Line 400- Who mentioned this; was it the HCW or the participant? Also note that at the end of each quote you should specify who said that quote

Line 401-Is it possible to get a quote indicating that the HCWs felts that the clients did not appreciate their work?

Line 404-405- Add whoever mentioned this quote at the end of the quote

Line 406-7-Where is the quote to sow that the HCWs reported a greater understanding of the clients’ feelings and their needs?

Line 410-13- Same comment as above

Line 415-18- Same comment as above

Line 419-23-Is it possible for the authors to back this statement up with a quote

Line 424-7-Could we get a quote from the HCWs to reflect this and also one from the PMTCT clients

Line 429- What does corrected information about HIV and ART mean?

Line 435-37-Include the participant who mentioned this quote at the end of the quote

General comment for the Results section: it’s usually appropriate to report results concurrently where mixed methods is used. Could the authors explore that possibility?

Discussion

Line 442-445- This is what the authors set out to find out before the intervention was implemented. So after the intervention what did the authors find out?

-Also the authors should be able to make the introductory section in the discussion section in line with the objectives of the study reporting significant findings in the first paragraph of the discussion section before going ahead to discuss each of the results in detail in the later paragraphs of this section

Line 449-451-This was the finding of the study but what does it imply? It would also be appropriate to give other studies with similar or dissimilar results when you are discussing your findings

Line 459-460-the authors should provide a reference for this statement

Line 463-465- What brought about this? Did it have any effect on the key findings of the study, if so then explain, giving other studies with the same or different findings

Line 492-Is 318 a reference or not? If it’s a reference, please correct it because reference 318 does not exist in the reference section

Line 492-497- Long sentence. Consider fragmenting it

-Also what makes the authors think that CSC is not an ideal mechanism to address systemic and interpersonal issues? Are there studies that suggest this? Please clarify and give a reference

7. PLOS authors have the option to publish the peer review history of their article (what does this mean?). If published, this will include your full peer review and any attached files.

Reviewer #4: No

Reviewer #5: **Yes: **Christine Aanyu

---

## [Author Response · Author response to Decision Letter 1]

21 Jul 2021

Thank you for your additional comments on the manuscript. A revised manuscript and response to reviewers has been submitted.

---

## [Editor Report · Decision Letter 2]

26 Jul 2021

Evaluating the effect of a Community Score Card among pregnant and breastfeeding women living with HIV in two districts in Malawi

PONE-D-20-35790R2

Dear Dr. Kays,

We’re pleased to inform you that your manuscript has been judged scientifically suitable for publication and will be formally accepted for publication once it meets all outstanding technical requirements.

Kind regards,

Hannah Tappis, DrPH, MPH

Academic Editor

PLOS ONE

---

## [Editor Report · Acceptance letter]

2 Aug 2021

PONE-D-20-35790R2 

Evaluating the effect of a Community Score Card among pregnant and breastfeeding women living with HIV in two districts in Malawi. 

Dear Dr. Kays:

I'm pleased to inform you that your manuscript has been deemed suitable for publication in PLOS ONE. Congratulations! Your manuscript is now with our production department. 

Kind regards, 

on behalf of

Dr. Hannah Tappis 

Academic Editor

PLOS ONE